# Beyond the Resistome: Molecular Insights, Emerging Therapies, and Environmental Drivers of Antibiotic Resistance

**DOI:** 10.3390/antibiotics14100995

**Published:** 2025-10-04

**Authors:** Nada M. Nass, Kawther A. Zaher

**Affiliations:** 1Department of Biological Sciences, Faculty of Science, King Abdulaziz University, Jeddah 21589, Saudi Arabia; nmnass@kau.edu.sa; 2Immunology Unit, King Fahd Medical Research Center, King Abdulaziz University, Jeddah 21589, Saudi Arabia; 3Department of Medical Laboratory Sciences, Faculty of Applied Medical Sciences, King Abdulaziz University, Jeddah 21589, Saudi Arabia

**Keywords:** antibiotic resistance, environmental resistome, wastewater, mobile genetic elements, *mcr*, biofilms, BL/BLI, CRISPR, One Health

## Abstract

Antibiotic resistance remains one of the most formidable challenges to modern medicine, threatening to outpace therapeutic innovation and undermine decades of clinical progress. While resistance was once viewed narrowly as a clinical phenomenon, it is now understood as the outcome of complex ecological and molecular interactions that span soil, water, agriculture, animals, and humans. Environmental reservoirs act as silent incubators of resistance genes, with horizontal gene transfer and stress-induced mutagenesis fueling their evolution and dissemination. At the molecular level, advances in genomics, structural biology, and systems microbiology have revealed intricate networks involving plasmid-mediated resistance, efflux pump regulation, integron dynamics, and CRISPR-Cas interactions, providing new insights into the adaptability of pathogens. Simultaneously, the environmental dimensions of resistance, from wastewater treatment plants and aquaculture to airborne dissemination, highlight the urgency of adopting a One Health framework. Yet, alongside this growing threat, novel therapeutic avenues are emerging. Innovative β-lactamase inhibitors, bacteriophage-based therapies, engineered lysins, antimicrobial peptides, and CRISPR-driven antimicrobials are redefining what constitutes an “antibiotic” in the twenty-first century. Furthermore, artificial intelligence and machine learning now accelerate drug discovery and resistance prediction, raising the possibility of precision-guided antimicrobial stewardship. This review synthesizes molecular insights, environmental drivers, and therapeutic innovations to present a comprehensive landscape of antibiotic resistance. By bridging ecological microbiology, molecular biology, and translational medicine, it outlines a roadmap for surveillance, prevention, and drug development while emphasizing the need for integrative policies to safeguard global health.

## 1. Introduction

Antibiotic resistance (AR) has shifted from a localized clinical challenge to a global health crisis of unprecedented scale. Projections warn that, if unaddressed, drug-resistant infections could account for 10 million deaths annually by 2050, surpassing cancer as the leading cause of mortality worldwide [1]. This alarming scenario is not solely the consequence of antibiotic misuse in medicine; rather, it reflects a complex molecular and ecological continuum that extends from environmental reservoirs to clinical pathogens.

The concept of the resistome has reshaped our understanding of AR. Far from being confined to hospitals, resistance determinants exist as an expansive genetic reservoir across soil, water, animals, and commensal microbes [2,3]. Modern metagenomics has revealed that many antibiotic resistance genes (ARGs) predate clinical antibiotic use by millions of years, having evolved in environmental bacteria as survival tools against naturally produced antimicrobial compounds [4,5]. Clinical multidrug resistance, therefore, often arises when selective pressures, such as antibiotic overuse in humans, agriculture, and aquaculture, mobilize these ancient genes into human pathogens [6].

At the molecular level, AR is driven by a spectrum of mechanisms, including chromosomal mutations, enzymatic drug inactivation, efflux pump overexpression, target modification, and horizontal gene transfer (HGT) [7]. Structural biology has provided unprecedented insights into the mechanisms of resistance enzymes. Cryo-electron microscopy and crystallography have elucidated how β-lactamases and carbapenemases adapt their catalytic sites, allowing even subtle amino acid substitutions to expand their substrate profiles [8,9]. Recent studies have mapped the flexibility of mobilized colistin resistance (MCR) proteins, a family of plasmid-borne phosphoethanolamine transferases, demonstrating how their activity can tolerate structural perturbations without compromising function, thereby enabling rapid global dissemination [10].

Environmental conditions magnify these molecular dynamics. Sub-inhibitory antibiotic concentrations, commonly found in wastewater treatment plants, agricultural soils, and aquaculture ponds, activate bacterial SOS responses that accelerate mutagenesis and prophage induction, thereby enhancing ARG mobilization [11,12]. Integrons, plasmids, and transposons serve as vehicles for this gene flow, while efflux pump regulators and small RNAs modulate bacterial fitness under stress [13]. The discovery of *mcr-9* and *mcr-10* variants in livestock and hospital sewage underscores the continuous emergence of novel resistance determinants at the environment–clinic interface [14].

An additional molecular layer is provided by CRISPR-Cas systems. In some bacteria, CRISPR arrays limit the acquisition of plasmids, thereby constraining the uptake of ARGs. In contrast, the absence of functional CRISPR machinery in specific pathogens facilitates the accumulation of multidrug resistance plasmids [15]. These dual roles highlight how genetic defense systems can both curb and promote the evolution of resistance. Beyond natural roles, engineered CRISPR-Cas antimicrobials now offer a futuristic therapeutic avenue by selectively excising resistance genes from bacterial genomes [16].

The persistence of resistant strains cannot be explained without considering the fitness costs associated with them. While resistance often reduces bacterial growth or competitiveness in antibiotic-free environments, compensatory mutations can restore or even enhance fitness [17]. This dynamic ensures that resistance traits remain entrenched in microbial populations long after antibiotic exposure ends.

From a translational perspective, these molecular insights are reshaping drug discovery, novel β-lactamase inhibitors (e.g., taniborbactam, zidebactam) target previously unmanageable carbapenemases [18]. Bacteriophage therapy and engineered endolysins exploit natural viral predation to dismantle bacterial defenses [19]. Antimicrobial peptides and synthetic biology platforms are providing new scaffolds less susceptible to traditional resistance pathways [20]. Meanwhile, artificial intelligence (AI) and machine learning tools are accelerating the identification of resistance escape routes and guiding the design of precision antibiotics [21].

Thus, the AR crisis must be understood as a continuum of molecular–ecological interactions. Environmental resistomes provide raw genetic material, molecular mechanisms ensure adaptability, and anthropogenic pressures accelerate dissemination. Combating AR will require integrated surveillance that combines clinical isolates with environmental sampling, encompassing sewage, soil, and animal microbiomes, utilizing metagenomics, functional assays, and AI-driven analytics [22,23]. Only by merging molecular biology, environmental science, and translational medicine can effective strategies be developed to safeguard global health.

This review will therefore explore three interwoven domains: molecular mechanisms of resistance, environmental reservoirs as amplifiers, and emerging therapies with prospects. The novelty of this review lies in its unified framework, which explicitly links molecular insights to both therapeutic and environmental strategies, while also incorporating artificial intelligence (AI), machine learning (ML), and One Health perspectives into a single roadmap. By integrating these dimensions, the review moves ‘beyond the resistome,’ advancing both fundamental understanding and translational innovation in antimicrobial resistance management.

## 2. The Molecular Resistome: Mechanisms and Evolutionary Fitness

The term resistome encompasses all antibiotic resistance genes (ARGs), including those intrinsic to bacterial genomes, those acquired via horizontal gene transfer (HGT), and cryptic determinants with the potential to evolve into active resistance mechanisms [24]. At the molecular level, the resistome is not static; it is an evolving network shaped by mutation, selection, and gene exchange across microbial communities. This dynamic nature explains why resistance has been detected even in environments historically untouched by modern antibiotics, such as permafrost and remote caves. ARGs have also been detected in cold, remote, and isolated environments, reinforcing the notion of an extensive environmental resistome. For example, a 2024 study on a Central Asian Mountain glacier recovered 944 ARGs across 22 antibiotic classes, with 633 shared across glacier layers, suggesting the deep penetration of antimicrobials, even into ice cores. Similarly, studies of glacier debris and meltwater in China have reported high frequencies of *blaCTX-M*, *tetA*, *sulI*, and additional ARGs in non-polar glacial environments. A global soil survey also revealed that ARG abundance peaks in high-latitude cold and boreal forests, driven partly by mobile genetic elements and climatic seasonality [25,26].

### 2.1. Chromosomal Mutations and Stress-Induced Variability

Chromosomal mutations remain a cornerstone of resistance evolution. Single-nucleotide polymorphisms can alter drug-binding sites, as exemplified by fluoroquinolone resistance through mutations in *gyrA* and *parC* [27]. Similarly, mutations in ribosomal RNA confer resistance to macrolides and aminoglycosides [28]. Beyond direct target modification, antibiotic exposure induces stress responses, such as the SOS regulon, a bacterial DNA-damage repair system that promotes mutagenesis and facilitates the mobilization of genetic elements [29]. Sub-inhibitory antibiotic concentrations, commonly detected in wastewater and soils, amplify this effect by promoting DNA damage repair pathways and recombination, thereby accelerating adaptive evolution [30].

Recent studies have highlighted how low-level β-lactam exposure enhances integron recombination, enabling the capture and shuffling of gene cassettes carrying ARGs [31]. The ecological implication is profound: resistance can emerge and stabilize in microbial communities even where antibiotic levels are far below therapeutic thresholds.

### 2.2. Plasmids and Mobile Genetic Elements

Plasmids remain the most critical vehicles for the dissemination of ARGs. Multi-resistance plasmids can carry genes for β-lactamases, aminoglycoside-modifying enzymes, and efflux systems simultaneously, conferring survival advantages under diverse antibiotic exposures [32]. Notably, plasmids do not evolve in isolation; compensatory mutations in both plasmids and host chromosomes can significantly reduce their fitness costs, allowing for stable persistence even in the absence of antibiotics [33].

Meta-analyses of plasmid-host interactions reveal that fitness penalties are not universal; some plasmids even enhance host competitiveness by encoding stress-tolerance genes [34]. The discovery of *mcr-9* and *mcr-10* on self-transmissible plasmids underscores the role of horizontal transfer in the global spread of colistin resistance [35]. Furthermore, long-read sequencing has revealed complex hybrid plasmids harboring virulence factors alongside ARGs, creating hyper-virulent and multi-resistant pathogens such as *Klebsiella pneumoniae* [36].

Mobile elements extend beyond plasmids. Integrative conjugative elements (ICEs), transposons, and insertion sequences (ISs) facilitate the mobilization of ARGs across bacterial species. Insertion sequences can activate silent resistance genes by providing promoters or disrupting repressor sequences [37]. CRISPR-associated phages and engineered CRISPR systems have been shown to influence ARG dynamics, either by mobilizing resistance determinants or by selectively degrading them in bacterial hosts [38]. Figure 1 shows a summary of the mechanism of drug resistance.

### 2.3. Efflux Pumps and Regulatory Networks

Efflux pumps, especially those of the RND (resistance-nodulation-division) family, expel structurally diverse antibiotics, including fluoroquinolones, tetracyclines, and carbapenems [39]. At the molecular level, the overexpression of efflux pumps results from mutations in local repressors (e.g., *mexR* in *Pseudomonas aeruginosa*) or global regulators, such as *mar*A and *sox*S, in *Escherichia coli* [40].

Recent work using transcriptomics and proteomics has revealed that efflux pumps are part of broader stress-response circuits, often co-regulated with oxidative stress defenses and biofilm formation [41]. This coupling enhances bacterial survival not only against antibiotics but also against host immune defenses, underscoring their dual role in resistance and virulence. Moreover, small RNAs (sRNAs) have emerged as fine-tuners of efflux pump activity, adding a post-transcriptional regulatory layer [42]. Targeting these regulatory nodes with small-molecule inhibitors represents an innovative therapeutic strategy currently under investigation.

### 2.4. Integrons and Gene Cassettes

Integrons serve as natural gene capture and expression systems, facilitating the dissemination of ARGs. They can incorporate diverse cassettes, ranging from β-lactamases to aminoglycoside-modifying enzymes, and their expression is enhanced under stress-induced SOS activation (DNA damage-triggered induction of error-prone repair pathways [43].

Recent metagenomic surveys demonstrate that class 1 integrons remain the dominant integron type in clinical and environmental isolates, particularly in wastewater treatment plants (Table 1) [44]. Alarming trends include the rise in integrons carrying carbapenemase genes (*blaNDM*, *blaKPC*) and co-localization with heavy-metal resistance determinants, suggesting co-selection in polluted environments [45].

### 2.5. CRISPR-Cas Systems: Dual Roles in Resistance

CRISPR-Cas systems, once considered barriers to HGT, now occupy a paradoxical position in the resistome. In some pathogens, the absence of CRISPR arrays correlates with a higher acquisition of multidrug resistance plasmids, suggesting that CRISPR can act as a gatekeeper against the influx of ARGs [51]. Conversely, in environmental bacteria, CRISPR-Cas elements can promote genomic plasticity by stimulating recombination events [45].

From a therapeutic standpoint, engineered CRISPR-Cas systems offer the possibility of selectively excising ARGs from pathogens. Proof-of-concept studies have shown plasmid curing and targeted ARG knockouts in *Enterobacteriaceae* and *Staphylococcus aureus* [52]. While clinical translation remains challenging, CRISPR antimicrobials represent a powerful molecular tool for reshaping resistomes.

### 2.6. Fitness Costs and Compensatory Evolution

Resistance mechanisms often come with trade-offs. For example, target-site mutations may reduce enzyme efficiency, and the overexpression of efflux pumps can burden metabolic resources [53]. However, bacteria frequently acquire compensatory mutations that mitigate these costs, allowing resistant strains to persist in the long term [54]. In some cases, plasmids carrying ARGs also harbor genes that restore metabolic balance, effectively neutralizing fitness penalties [55].

Recent long-term evolution experiments demonstrate that resistance traits can become “cost-free” within a few hundred generations, particularly when multiple selective pressures (such as antibiotics, biocides, and heavy metals) interact [56]. This compensatory evolution explains why withdrawal of specific antibiotics (e.g., fluoroquinolones) does not always reverse resistance trends in clinical settings [57]. As summarized in Figure 2, mobile genetic elements (MGEs) act as the hub that links target-site modification to other resistance modules, explaining their frequent co-occurrence on the same replicon.

## 3. The Resistome as a Reservoir for Future Resistance

One of the most concerning insights from molecular studies is that the resistome contains cryptic ARGs—genes not yet mobilized into pathogens but with the potential to become clinically relevant under selective pressure [58]. Functional metagenomics has uncovered dozens of novel β-lactamases, tetracycline-inactivating enzymes, and efflux systems in soil and aquatic microbiomes [59]. Many of these ARGs can be mobilized by integrons or plasmids, raising the possibility that future clinical resistance will emerge from today’s environmental reservoirs.

Computational pipelines using machine learning now predict ARG mobilization potential by analyzing sequence features such as proximity to mobile elements and GC-content compatibility with human pathogens [60]. Such predictive models represent the next frontier in surveillance, providing early warnings before novel ARGs enter clinical circulation (Table 2).

## 4. Emerging Therapies and Molecularly Informed Interventions

The accelerating antibiotic resistance (AR) crisis compels the development of therapeutic strategies that transcend traditional antibiotics. Conventional antimicrobials are increasingly undermined by enzymatic inactivation, efflux pump upregulation, and the rapid dissemination of resistance determinants across plasmids, integrons, and transposons. Recent progress highlights a multi-pronged approach that combines drug innovation, nanotechnology-driven disinfection, bio-inspired molecules, and molecular diagnostics to suppress the emergence, persistence, and transmission of antibiotic resistance genes (ARGs). Therapies were directly mapped onto dominant resistance mechanisms in Figure 3, highlighting where redesign (e.g., macolacin vs. mcr) and potentiation (e.g., BL/BLI, efflux/OM adjuvants) compress evolutionary escape. Two complementary strategies dominate: (i) resistance-evasive redesign of legacy scaffolds to restore target engagement under altered chemistries (e.g., PEtN-modified lipid A in mcr contexts), and (ii) potentiation of core antibiotics with adjuvants that neutralize resistance enzymes or recondition bacterial physiology (efflux, OM permeability, biofilm tolerance).

### 4.1. Electrochemical and Catalytic Strategies

Electrochemical advanced oxidation processes (EAOPs) represent promising adjunct technologies for wastewater treatment and the degradation of antibiotics. In one study, Ti/PbO‚ÇÇ-based electrodes achieved complete chloramphenicol (CAP) degradation with 100% total organic carbon (TOC) removal under optimized conditions (15.0 mA/cm^2^, pH 5, 0.125 mol/L electrolyte). Significantly, incomplete degradation promoted an increase in ARG abundance (*intI1*, *cmlA*, *cmle3*, *cata2*) in river sediments, while complete mineralization prevented ARG proliferation [56]. These findings emphasize that partial antibiotic degradation may paradoxically increase ARG prevalence, underscoring the need for technologies that ensure complete mineralization before effluent discharge.

Catalytic nanomaterials further expand this paradigm. Nanomaterial-enhanced hybrid disinfection using photocatalysts and ultrasonication generated reactive oxygen species (ROS) that eliminated multidrug-resistant (MDR) bacteria while degrading ARGs in wastewater. The synergy between nanocomposites and natural biocides increased disinfection efficiency compared with chlorination and UV alone [57]. This suggests that nanotechnology-driven oxidation processes may both inactivate resistant pathogens and fragment extracellular ARGs, reducing genetic pollution.

### 4.2. Nanomaterials and the Plastisphere

Plastic pollution introduces another layer of complexity. Micro- and nanoplastics (MNPs) provide durable substrates for microbial colonization, creating the so-called “plastisphere.” Microbial biofilms on MNPs enrich ARGs such as *blaTEM*, *qnrS*, *sulI*, and *ermB*, which persist even after conventional wastewater treatment. Weathered or aged plastics exacerbate ARG proliferation by increasing the surface adsorption and desorption of antibiotics, metals, and pathogens [58]. This confirms that plastics act as ARG vectors, promoting horizontal gene transfer within aquatic ecosystems.

Emerging mitigation strategies target the interface between the plastisphere and ARG. Green filtration systems for nitrate-, pesticide-, and antibiotic-polluted groundwater effectively reduced ARG loads while restoring water quality [59]. Soil bioremediation approaches, including biochar amendments, have also been shown to suppress the persistence of ARGs in urban soils [60]. Together, these findings suggest that both plastic pollution control and bioremediation are critical ARG mitigation strategies.

### 4.3. Bio-Inspired Natural Compounds

Plant-derived bioactives show promise in combating resistant pathogens. Andrographolide, quercetin, and rutin inhibited Aeromonas hydrophila biofilms and suppressed ARG-related gene expression, acting synergistically to reduce quorum sensing and oxidative stress pathways [61]. Their dual role as antibiofilm and anti-ARG agents positions natural products as resistance-modulating adjuvants.

At the microbial level, a naturally inspired colistin congener, macolacin, was identified from a divergent biosynthetic gene cluster. The macolacin class exemplifies pharmacophore redesign to retain activity against PEtN-modified lipid A in mcr-positive backgrounds. Unlike colistin, macolacin remained effective against *mcr-1*-positive Gram-negative pathogens, including extensively drug-resistant *Acinetobacter baumannii* and intrinsically colistin-resistant *Neisseria gonorrhoeae*. In murine infection models, biphenyl macrolactin analogs demonstrated potent in vivo efficacy, reviving polymyxin scaffolds as viable therapies [62]. This highlights the untapped evolutionary solutions within microbial metabolite diversity that can bypass resistance barriers.

### 4.4. Molecular Surveillance and CRISPR-Based Tools

In parallel with therapy, the early detection and surveillance of ARGs are critical. A portable CRISPR-Cas12a toolbox was developed that detected ARG markers (*blaCTX-M-15*, *floR*) at attomolar sensitivity. This platform combined CRISPR/Cas12a with PCR and lateral flow assays, enabling on-site identification of ARGs in clinical and environmental samples. Detection extended to intI1, confirming the prevalence of integron-associated resistance [66]. Such tools represent a significant leap toward field-deployable, low-cost ARG diagnostics, which are crucial for real-time One Health surveillance. CRISPR antimicrobials occupy the plasmid-vehicle axis in Figure 4, lowering baseline resistance by curing ARG plasmids.

CRISPR also plays a therapeutic role. Engineered CRISPR constructs, such as the VADER system, demonstrated targeted plasmid curing and ARG degradation in wastewater. By coupling conjugative delivery with CRISPR-Cas immunity, VADER achieved complete elimination of resistance plasmids in *E. coli* and *Pseudomonas aeruginosa* [64]. This approach illustrates how genome-editing technologies may transform environmental resistome management (Table 3).

### 4.5. Integrative Therapeutic Roadmap

The convergence of molecular and environmental strategies highlights the need for integrated interventions:
○At the environmental interface, advanced oxidation, nanomaterials, and green remediation must ensure the complete removal of antibiotics and ARGs before ecological release.○At the molecular interface, natural products (such as quercetin and macolacin) and engineered antimicrobials (including CRISPR and phages) provide new scaffolds to bypass resistance mechanisms.○At the surveillance interface: Portable CRISPR-Cas detection and metagenomic pipelines enable real-time ARG tracking across soil, water, and clinical isolates.

The synthesis of these innovations signals a shift from reactive to preventive resistance management, aligning with the One Health vision. Operationally, the layered approach in Figure 4 maps directly onto the integrated strategy, summarized as removing selection pressure, interrupting genetic vehicles, and instrumenting with data

## 5. Surveillance and One Health Approaches

The management of antibiotic resistance (AR) extends beyond drug discovery and treatment. Effective containment requires surveillance systems that monitor resistance in humans, animals, and the environment under a unified One Health framework. The global resistome is a dynamic and interconnected entity; therefore, continuous monitoring, early warning systems, and integrative policies are essential to curb the emergence and dissemination of resistance genes.

### 5.1. Environmental Surveillance of ARGs

Wastewater-based surveillance has emerged as a powerful tool to detect ARGs before they become established in clinical pathogens. Metagenomic analyses of sewage have successfully tracked extended-spectrum β-lactamases (*blaCTX-M*), carbapenemases (*blaNDM*, *blaKPC*), and mobile colistin resistance (*mcr*) genes across continents [84]. In fact, municipal wastewater often reflects community-level antimicrobial consumption, serving as a cost-effective mirror of public health. As summarized in Figure 5, clinical, veterinary, and environmental data streams are funneled to a central metagenomics/CRISPR analytics hub that produces decision-ready outputs for policy and stewardship.

Recent studies highlight wastewater treatment plants (WWTPs) as both ARG hotspots and surveillance nodes. Conventional treatment frequently fails to eliminate ARGs, with effluents containing high levels of extracellular DNA and mobile elements [66]. Advanced sequencing has revealed co-occurrence of ARGs with virulence genes, suggesting that resistant and more virulent strains may be selected in WWTPs [84]. Incorporating metagenomics, qPCR panels, and machine learning into WWTP surveillance could provide real-time risk assessment for emerging ARGs. Evidence synthesized here spans laboratory to pilot/field scales (including preprints in diagnostic platforms). Generalizability, cost, and real-world effectiveness, especially for CRISPR antimicrobials and advanced polishing trains, require further evaluation; we highlight these as priorities for translational studies. Figure 5 illustrates the data flow from source domains into a unified platform, enabling cross-sector situational awareness and early warning for emerging ARGs.

### 5.2. Clinical and Zoonotic Monitoring

Surveillance within hospitals remains critical. Genomic studies of *Klebsiella pneumoniae* and *Escherichia coli* clinical isolates have revealed the convergence of ARGs and virulence determinants, resulting in high-risk clones associated with prolonged outbreaks [85]. The detection of ESBL and carbapenemase producers is particularly concerning in intensive care units where immunocompromised patients are most vulnerable [86].

Zoonotic transmission adds another layer of complexity. ARGs identified in *Salmonella enterica serovars* from livestock and food chains often overlap with resistance determinants in clinical strains, highlighting the porous boundary between agricultural and human health resistomes [87]. Continuous monitoring of veterinary isolates, combined with traceability in food production, is therefore a cornerstone of One Health resistance control.

### 5.3. Mobile Genetic Elements as Surveillance Targets

Mobile genetic elements (MGEs), including plasmids, integrons, and transposons, serve as central conduits for ARG dissemination. A global genomic survey revealed that more than 60% of clinically relevant ARGs are mobilized by MGEs [88]. This finding underscores the importance of tracking not only resistant pathogens but also the genetic mechanisms responsible for the spread of resistance.

Portable tools such as CRISPR-based diagnostics are now being designed to detect MGE-associated ARGs in realtime. For example, Cas12a-based systems have been applied to monitor integron-associated *blaCTX-M* and *floR* genes in field settings [89]. Such platforms could provide low-cost, rapid ARG surveillance in both clinical and environmental settings, extending surveillance capacity to low-resource regions.

### 5.4. Phage and Resistome Surveillance

Bacteriophages, once considered passive carriers, are increasingly recognized as ARG vectors. Metavirome studies in WWTPs and river sediments have detected ARGs encoded on phage genomes [90]. This highlights the need to incorporate viral metagenomics into frameworks for monitoring ARGs. Phage surveillance can also guide the development of therapeutic phages while preventing the dissemination of inadvertent resistance.

### 5.5. Artificial Intelligence in ARG Prediction

Machine learning (ML) is revolutionizing ARG surveillance by predicting novel resistance determinants from environmental and clinical datasets. By analyzing genetic signatures such as GC content, codon usage, and association with MGEs, ML models can forecast the mobilization potential of previously uncharacterized genes [91]. These predictive tools offer a proactive approach, identifying resistance threats before they establish in pathogens.

Recent platforms combine ML with wastewater sequencing data to estimate local and regional resistance prevalence. This approach has already been piloted in Europe, enabling predictive mapping of β-lactam and colistin resistance [92]. When integrated into global health networks, AI-driven surveillance could provide early warning systems against upcoming resistance waves.

### 5.6. One Health Integration

The One Health framework recognizes the interdependence of human, animal, and environmental health in combating AR. Antibiotics used in agriculture select for ARGs that can transfer to human pathogens, while resistant bacteria from hospitals are discharged into wastewater, creating circular pathways of resistance exchange [93]. Policies must therefore integrate agricultural regulation, hospital stewardship, and environmental protection (Figure 5).

Promising initiatives include coordinated One Health surveillance hubs that link hospital laboratories, veterinary clinics, and environmental monitoring stations. These hubs generate shared resistome datasets, allowing cross-sectoral analysis of ARG emergence and dissemination [94]. Countries such as Sweden and the Netherlands have pioneered integrated resistance databases, serving as models for global adoption (Table 4).

### 5.7. Policy and Governance

Translating surveillance into action requires robust policy frameworks. Current gaps include underreporting in low- and middle-income countries, limited integration of environmental data, and fragmented governance across sectors [70]. Global agencies advocate for standardized metagenomic pipelines, shared databases, and harmonized reporting to enable cross-country comparisons.

Furthermore, surveillance data must inform policy interventions such as restrictions on non-therapeutic antibiotic use in livestock, incentives for antibiotic stewardship programs, and investment in wastewater treatment technologies. Without governance structures that link surveillance to regulatory enforcement, resistance monitoring risks becoming a descriptive exercise rather than a preventive tool.

## 6. Prospects and Future Directions

The ongoing battle against AR highlights the urgent need to integrate molecular insights, innovative therapies, and global policy into a cohesive roadmap. While substantial progress has been made in understanding resistance mechanisms and environmental drivers, the future of AR control depends on translating knowledge into sustainable interventions. The convergence of biotechnology, nanoscience, synthetic biology, and artificial intelligence promises to redefine the future of antimicrobial therapy and resistance prevention.

### 6.1. Next-Generation Antibiotics and Adjuvants

Traditional drug discovery pipelines remain slow, yet bio-inspired molecules are providing a new arsenal. The discovery of macolacin, an engineered colistin analog, demonstrated that redesigning natural scaffolds can circumvent resistance conferred by *mcr* genes [71]. Similarly, novel β-lactamase inhibitors such as taniborbactam and zidebactam expand the utility of carbapenems against multidrug-resistant Gram-negatives [73].

Future directions emphasize the combination of old drugs with new adjuvants, exploiting synergy to resensitize pathogens. Plant-derived bioactives like quercetin and rutin, shown to suppress ARG expression and biofilm formation, are being studied as resistance-modifying agents [74]. This adjuvant strategy aligns with the principle of resistance reversion, where natural molecules weaken bacterial defenses and restore the potency of antibiotics.

### 6.2. Phage Therapy and Engineered Enzybiotics

Bacteriophages represent one of the most promising future therapeutics. Phages can specifically target multidrug-resistant bacteria, leaving the commensal microbiota intact. Clinical phage applications are expanding, with successful compassionate-use cases reported against Acinetobacter baumannii and *Klebsiella pneumoniae* infections [75]. Beyond natural phages, engineered variants expressing CRISPR-Cas payloads or enhanced endolysins can directly degrade resistance genes or biofilm matrices [76].

Future work must focus on standardizing phage therapy trials, building regulatory frameworks, and establishing biobanks of phage cocktails targeting critical pathogens. Combining phages with antibiotics or nanoparticles could overcome both resistance and biofilm-associated tolerance.

### 6.3. CRISPR-Based Therapeutics and Diagnostics

Genome-editing technologies hold transformative potential. CRISPR-Cas antimicrobials, such as the VADER system, have already demonstrated plasmid curing and ARG elimination in wastewater settings [79]. As delivery systems improve, using phagemids, conjugative plasmids, or lipid nanoparticles, CRISPR antimicrobials may be applied directly to patients to excise ARGs from pathogens.

On the diagnostic front, portable CRISPR-based platforms are revolutionizing surveillance. Cas12a and Cas13 tools offer rapid, field-deployable detection of ARGs with attomolar sensitivity [92]. Future diagnostics will likely integrate these CRISPR platforms with microfluidics and smartphone-based readers, allowing global, low-cost surveillance across clinical and environmental settings.

### 6.4. Nanotechnology and Advanced Materials

Nanomaterials are set to play a dual role in the future of AR control: as therapeutics and as environmental remediators. Hybrid nanomaterials producing ROS have shown the ability to eliminate resistant bacteria while degrading extracellular ARGs [86]. Metal–organic frameworks (MOFs) and functionalized biochar are also being developed to adsorb antibiotics and ARGs in wastewater [81].

Looking ahead, nanotechnology will likely underpin precision antimicrobial delivery systems-nanocarriers that release drugs in response to infection-specific triggers, reducing off-target effects and limiting selective pressure on non-pathogenic bacteria.

### 6.5. Artificial Intelligence and Predictive Surveillance

Artificial intelligence (AI) and machine learning (ML) are becoming indispensable tools in combating AR. Deep-learning algorithms have already discovered novel antimicrobials such as halicin and macolacin [93]. More recently, ML pipelines have been applied to metagenomic surveillance, predicting ARG mobilization potential and forecasting resistance prevalence across regions [94].

In the future, AI-driven drug design may reduce development time by decades, while predictive ARG mapping will enable proactive interventions in communities at the highest risk of resistance outbreaks. Integrating AI into global surveillance databases will be crucial for developing timely, evidence-based policies.

### 6.6. One Health Policy and Global Cooperation

The One Health framework remains the cornerstone of future AR strategies. Resistance surveillance must integrate human, animal, and environmental data into unified networks, utilizing standardized metagenomic pipelines and real-time dashboards that are accessible across borders [95].

Global agencies are now emphasizing investments in wastewater treatment upgrades, agricultural regulation, and incentives for antibiotic stewardship [96]. Without coordinated policy enforcement, even the most innovative therapies risk being undermined by the uncontrolled dissemination of ARGs in the environment.

## 7. Vision: Preventive Resistance Management

While Section 6 detailed emerging scientific and technological prospects, this section builds upon those insights to propose a broader preventive framework. Rather than repeating mechanisms, the focus here is on how these innovations can be strategically integrated into One Health governance and global resistance management.

The ultimate prospect in combating antibiotic resistance lies in shifting from a reactive to a preventive paradigm. Rather than waiting for multidrug-resistant infections to emerge and spread, prevention requires an integrated strategy that combines innovation, ecological containment, and governance.

This preventive vision rests on four pillars:Molecular precision tools: Approaches such as CRISPR-based antimicrobials and AI-designed therapeutics offer the capacity to preemptively block the mobilization and spread of resistance genes.Environmental containment: Nanotechnology, biochar amendments, and green filtration strategies reduce the recycling of ARGs in wastewater, soils, and aquatic ecosystems.Surveillance integration: Global resistome monitoring—linking clinical, veterinary, and environmental data through metagenomics and portable CRISPR diagnostics—enables early detection and targeted interventions.Policy enforcement: Coordinated international frameworks are essential to align antibiotic stewardship with sustainability and One Health principles, ensuring that scientific progress is effectively implemented in practice.

If implemented together, these measures could contain resistance before it reaches catastrophic levels. By harnessing the synergies of molecular innovation, ecological management, and global cooperation, the coming decades could represent not only a turning point in the fight against antimicrobial resistance but also a model for preventive global health governance.

As summarized in Figure 5 and Figure 6, recent milestones illustrate this trajectory: from global sewage metagenomics baselines (2019) and predictive links to clinical resistance (2020), through the GLASS dashboard/report updates (2022), to CRISPR plasmid-curing demonstrations (2023), portable Cas12a ARG assays (2024), and field evaluation of green groundwater filtration technologies (2025). Collectively, these advances enable earlier detection, targeted engineering controls, and informed One Health policy—marking the transition from reaction to prevention.

## 8. Conclusions

AR remains one of the most pressing global health challenges of the 21st century. What began as an evolutionary adaptation of environmental microbes has now escalated into a crisis threatening every dimension of human, animal, and environmental health. This review highlighted the molecular underpinnings of resistance, encompassing chromosomal mutations, enzymatic drug inactivation, efflux pump regulation, and the dissemination of mobile genetic elements. These mechanisms, once confined to ecological niches, have been amplified by human activity, transforming the environment into a global reservoir of ARGs [97].

The role of environmental interfaces, particularly wastewater treatment plants, soils, and aquatic systems, underscores the ecological entrenchment of resistance. ARGs are no longer restricted to pathogenic bacteria; they persist in biofilms, microplastics, and even biodegradable materials, often enhanced by co-selection pressures from metals, pesticides, and disinfectants [93]. Thus, tackling AR requires addressing not only clinical misuse of antibiotics but also environmental contamination, agricultural overuse, and waste mismanagement.

On the therapeutic front, the pipeline is cautiously optimistic. Novel antibiotics, such as macolacin, demonstrate that the structural redesign of natural scaffolds can overcome last-line resistance barriers [98]. Phage therapy and engineered enzybiotics are re-emerging as precision tools against multidrug-resistant bacteria, offering pathogen-specific alternatives with fewer off-target effects [77]. At the same time, CRISPR-Cas systems hold dual promise, both as antimicrobials capable of excising resistance determinants and as diagnostics for the rapid, low-cost detection of ARGs in clinical and environmental settings [80]. Nanotechnology adds yet another dimension, enabling ROS-driven bacterial killing, targeted drug delivery, and adsorption of ARGs in wastewater streams [97].

Yet, innovation without surveillance and governance cannot achieve sustainable control. The One Health framework is central to future success, emphasizing the interconnectedness of humans, animals, and the environment. Wastewater metagenomics, AI-powered ARG prediction, and global resistance databases are pioneering real-time surveillance systems [93]. For these technologies to have impact, they must be embedded within robust policy frameworks, supported by international collaboration and equitable resource distribution [99].

Looking forward, the most significant challenge lies in shifting from reactive crisis management to preventive resistance governance. This will require:
○Global surveillance integration, linking clinical, agricultural, and environmental ARG data.○Sustainable therapeutic innovation, balancing new antibiotics with phages, CRISPR-based tools, and natural adjuvants.○Environmental remediation, ensuring wastewater treatment, soil bioremediation, and plastic pollution control, reduces ARG dissemination.○Policy enforcement, limiting unnecessary antibiotic use, and incentivizing stewardship across sectors.

The prospect is clear: antibiotic resistance cannot be eliminated, but it can be contained, managed, and mitigated through interdisciplinary action. By uniting molecular science, environmental stewardship, biotechnology innovation, and One Health policy, the future of AR management can shift from inevitability to opportunity. Notably, this review diverges from prior literature by providing a comprehensive synthesis that integrates molecular insights with therapeutic and environmental strategies, and by incorporating AI/ML-driven surveillance into the One Health framework. Such an integrative perspective defines the novelty of this work and underscores its relevance to developing preventive resistance management strategies. This comprehensive approach will ensure that antibiotics remain effective for generations to come, preserving their role as the foundation of modern medicine.

## Figures and Tables

**Figure 1 antibiotics-14-00995-f001:**
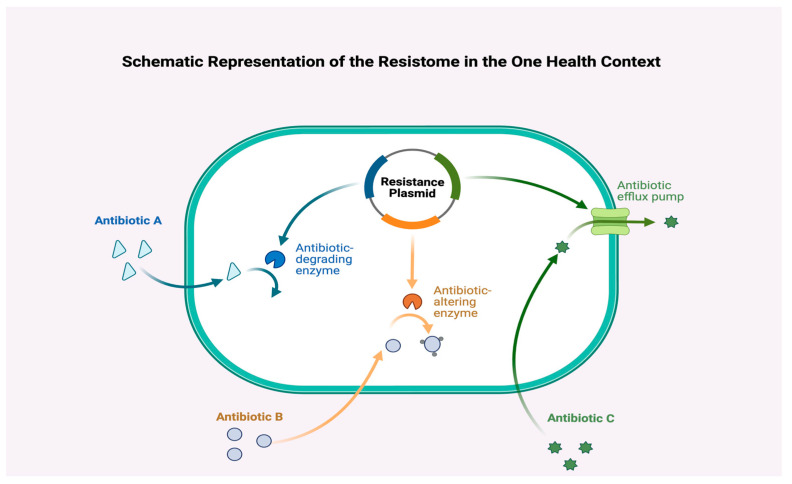
Plasmid-encoded resistance mechanisms inside a Gram-negative bacterial cell. A resistance plasmid concurrently encodes (i) an antibiotic-degrading enzyme (blue path; e.g., β-lactamases), (ii) an antibiotic-altering enzyme (orange path; e.g., aminoglycoside acetyl/adenyl/phospho-transferases), and (iii) an efflux pump (green path; e.g., RND systems). Together, these functions reduce intracellular antibiotic exposure and produce multidrug phenotypes under a single mobile element.

**Figure 2 antibiotics-14-00995-f002:**
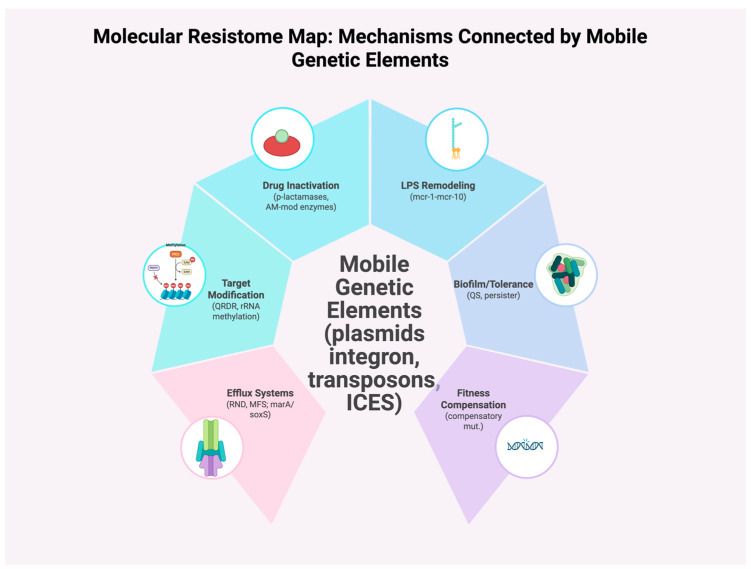
Molecular resistome map: mechanisms connected by mobile genetic elements. A central hub denotes mobile genetic elements (MGEs), including plasmids, integrons, and transposons, that mobilize and co-assemble resistance modules. Spokes summarize the principal strategies: drug inactivation (β-lactamases, aminoglycoside-modifying enzymes), target modification (quinolone resistance-determining region mutations, rRNA methylation), efflux systems (resistance-nodulation-division and major facilitator superfamily transporters; regulation by ***marA***/***soxS***), lipopolysaccharide (LPS) remodeling (***mcr-1*** to ***mcr-10***), biofilm/tolerance (quorum sensing, persisters), and fitness compensation (compensatory mutations) that stabilize resistance genes in bacterial populations.To avoid confusion, it is essential to distinguish between mechanisms already active in clinical and environmental contexts (as described above) and latent or cryptic resistance determinants that may only become mobilized under future selective pressures. The latter are the focus of the following section.

**Figure 3 antibiotics-14-00995-f003:**
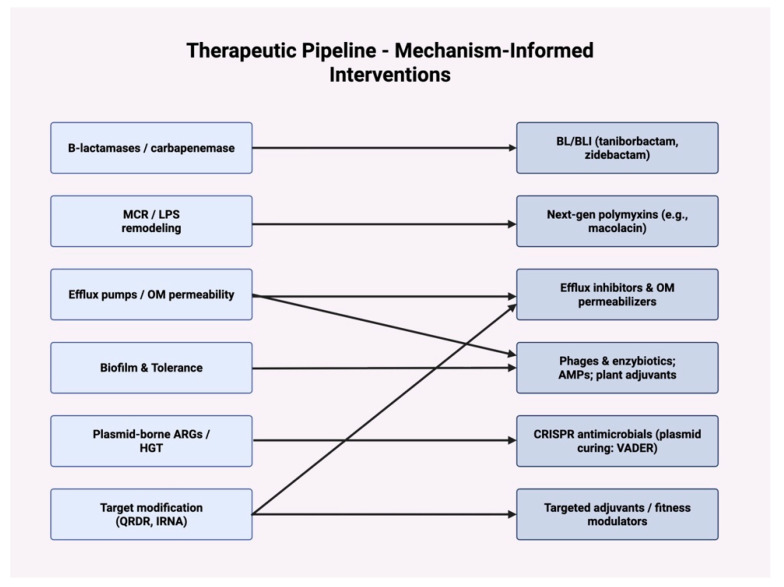
Therapeutic pipeline-mechanism-informed interventions. Left column lists key resistance mechanisms (β-lactamases/carbapenemases; mcr-mediated LPS remodeling; efflux/outer-membrane permeability; biofilm/tolerance; plasmid-borne ARGs/HGT; target modification, QRDR, rRNA). Right column aligns principal interventions (BL/BLI pairs, e.g., taniborbactam, zidebactam; next-generation polymyxins, e.g., macolacin; efflux inhibitors and OM permeabilizers; phages & enzybiotics; AMPs and plant adjuvants; CRISPR antimicrobials for plasmid curing, e.g., VADER; targeted adjuvants/fitness modulators). Arrows indicate primary pairings; combination therapy across classes is often required. Abbreviations: BL/BLI, β-lactam/β-lactamase inhibitor; OM, outer membrane; AMPs, antimicrobial peptides; HGT, horizontal gene transfer; QRDR, quinolone resistance-determining region.

**Figure 4 antibiotics-14-00995-f004:**
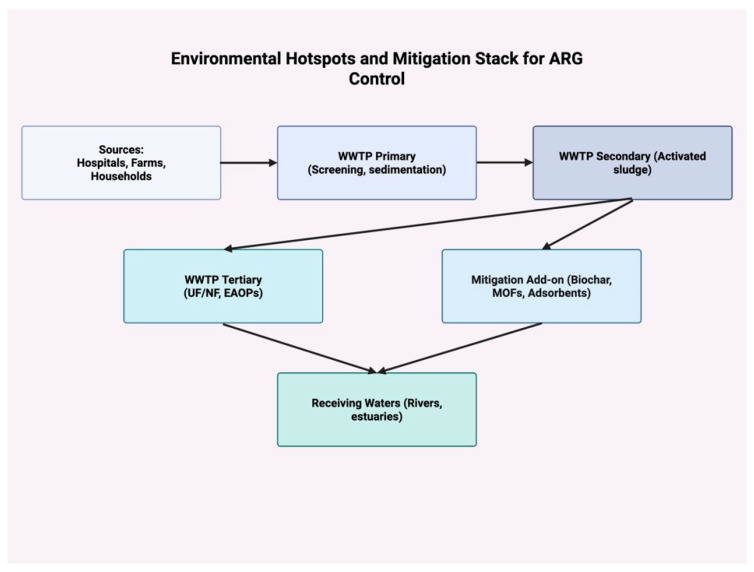
Environmental hotspots and mitigation stack for ARG control. Flow diagram from Sources (hospitals, farms, households) through WWTP Primary (screening/sedimentation) and WWTP Secondary (activated sludge) to two polishing pathways: WWTP Tertiary (UF/NF, EAOPs) and Mitigation Add-ons (biochar, MOFs, adsorbents). Arrows indicate complementary upgrade routes converging on Receiving Waters (rivers/estuaries). The concept emphasizes that secondary treatment may leave extracellular DNA and ARG carriers, whereas tertiary polishing and sorbent add-ons reduce antibiotic residues and ARG loads before discharge. Abbreviations: WWTP, wastewater treatment plant; UF/NF, ultrafiltration/nanofiltration; EAOPs, electrochemical advanced oxidation processes; MOFs, metal–organic frameworks; ARG, antibiotic resistance gene.

**Figure 5 antibiotics-14-00995-f005:**
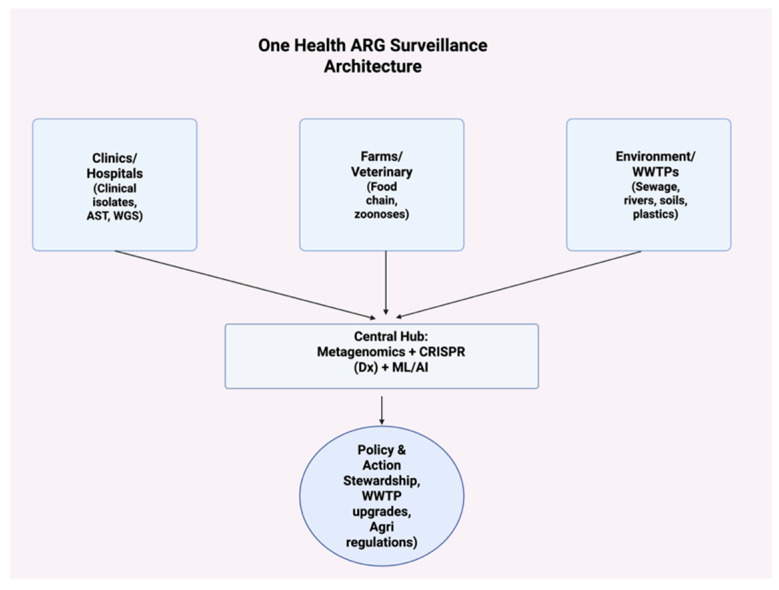
One Health ARG surveillance architecture. Streams from Clinics/Hospitals (clinical isolates, AST, WGS), Farms/Veterinary (food-chain, zoonoses), and the Environment/WWTPs (sewage, rivers, soils, plastics) converge on a central hub that integrates metagenomics with CRISPR-based diagnostics and ML/AI analytics; outputs inform Policy & Action (stewardship, WWTP upgrades, agricultural regulations). Abbreviations: ARG, antibiotic resistance gene; AST, antimicrobial susceptibility testing; WGS, whole-genome sequencing; WWTP, wastewater treatment plant; ML/AI, machine learning/artificial intelligence.

**Figure 6 antibiotics-14-00995-f006:**
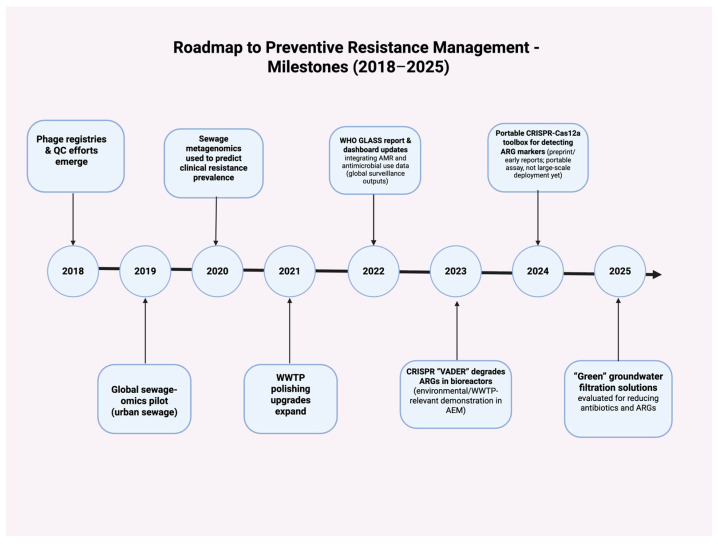
Roadmap to Preventive Resistance Management Milestones (2018–2025).

**Table 1 antibiotics-14-00995-t001:** Molecular mechanisms of antibiotic resistance, representative genes/targets, and clinical relevance.

Mechanism	Molecular Example(s)	Mobile Elements/ Regulation	Clinical Relevance	Key Refs
Drug inactivation (β-lactams)	Class A/C/D ESBLs; NDM/VIM/IMP metallo-β-lactamases	Plasmids, integrons; SOS-induced cassette exchange	Carbapenem and cephalosporin failure in Enterobacterales	[8,9,24,39,43,44,45]
Target protection/modification	QRDR mutations in *gyrA*/*parC* (FQs); rRNA methylation (erm)	IS activation, transposons	Fluoroquinolone and macrolide resistance across *E. coli*, *S. aureus*	[27,28,37]
Efflux overexpression	RND pumps (MexAB-OprM, AcrAB-TolC); regulators marA/soxS	sRNAs; global stress circuits	Broad MDR in *P. aeruginosa*, *E. coli*	[39,40,41,42]
Outer membrane/LPS remodeling	*mcr-1*–*mcr-10* phosphoethanolamine transferases	Conjugative plasmids	Colistin failure; spread from agriculture→clinic	[10,35]
Bypass/synthesis	Altered PBPs; folate pathway detours	ICEs, integrons	β-lactam and TMP-SMX compromised	[43,44]
Biofilm-linked tolerance	QS, matrix genes; oxidative stress reprogramming	—	Persisters; device infections	[41,42]
Fitness compensation	Compensatory chromosomal/plasmid mutations	—	Persistence of ARGs without antibiotic pressure	[17,34,46,47,48,49,50]

**Table 2 antibiotics-14-00995-t002:** Environmental ARG hotspots and mitigation technologies, accompanied by recent performance data.

Hotspot	Typical ARGs/Drivers	Key Findings (Performance)	Technology/Approach	Key Refs
WWTP secondary effluent	*blaKPC*, *blaNDM*, *mcr*, *intI1*; *sub-MICs*	Conventional steps often leave extracellular ARGs	Add tertiary barrier + EAOP/nanofiltration	[61,62,63]
WWTP tertiary (UF/NF)	Free-floating ARGs	UF → ARG passage; NF removed >99.9% total bacteria & ARGs in effluent	Nanofiltration polishing	[62,63]
Anaerobic digestion	*tet*, *sul*; co-selection	ARG persistence unless optimized; biochar/consortia recommended	Process optimization + amendments	[63]
Groundwater (nitrate/pesticides/antibiotics)	Mixed ARGs; co-pollutants	“Green” filtration reduced contaminants and ARG loads	Biofiltration + sorbents	[64]
Urban/agricultural soils	Metals, pesticides; manure	Biochar & soil management lowered ARG incidence	Bioremediation	[65]
Plastisphere (micro/nanoplastics)	*blaTEM*, *qnrS*, *sulI*, *ermB*; aged surfaces	Biofilms on MNPs enrich ARGs despite treatment	Plastic pollution control + targeted removal	[63]
Rivers/estuaries	Nutrients, metals	Nonlinear links between nutrients/metals and ARG abundance	Source control + adsorbents/MOFs	[66,67,68,69,70]
Bioplastics (PLA/PHB)	Proliferation under anaerobic degradation	Enhanced ARG proliferation during biodegradation	Life-cycle scrutiny & controls	[68]

**Table 3 antibiotics-14-00995-t003:** Emerging therapies and molecularly informed interventions.

Modality	Molecular Target/Concept	Status/Notes	Key Refs
Macolacin (polymyxin congener)	Active vs. *mcr*-positive GNB	In vivo efficacy in murine models	[62,71,72]
New BL/BLI pairs (e.g., taniborbactam, zidebactam)	broaden carbapenemase coverage	Late stage/updates 2023	[18,73]
Antimicrobial peptides & plant bioactives (quercetin, rutin, andrographolide)	Anti-biofilm, QS, oxidative pathways; ARG expression down	Synergistic adjuvants	[20,66,74]
Phage therapy/engineered lysins	Pathogen-specific lysis; biofilm disruption	Clinical case expansion; standardization needed	[19,75,76,77]
CRISPR antimicrobials (e.g., VADER)	Plasmid curing/ARG excision	Demonstrated in bioreactors/WWTP contexts	[78,79,80]
Nanomaterial hybrid disinfection	ROS-based kill; extracellular DNA fragmentation	Outperforms UV/chlorination; pair with polishing	[57,81,82,83]

**Table 4 antibiotics-14-00995-t004:** One Health Surveillance Toolkit and Deployment Characteristics.

Tool	What It Detects	Deployment	LOD/Speed	Notes	Key Refs
Sewage metagenomics (shotgun)	Full ARGome + pathogens	WWTP influent/effluent	High depth; batch (hours–days)	Community-level early warning	[69,84,95]
CRISPR-Cas diagnostics (Cas12a)	Specific ARGs (e.g., *blaCTX-M-15*, *floR*, *intI1*)	Fieldable, lateral-flow	Attomolar; minutes–<1 h	Low-cost, portable	[66,80,92]
Viral metagenomics	Phage-borne ARGs	Rivers, WWTPs	Batch	Complements bacterial WGS	[68]
ML/AI prevalence prediction	ARG mobilization, regional risk	Dashboard	Real-time once trained	Decision support for policy	[69,78,79]

## Data Availability

No new data were created or analyzed in this review; all information is available in the cited publications and public repositories referenced herein. Further inquiries can be directed to the corresponding author.

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
