# Peer review of "Beyond the Resistome: Molecular Insights, Emerging Therapies, and Environmental Drivers of Antibiotic Resistance"

_antibiotics, 2025, doi:10.3390/antibiotics14100995_

Round 1

Reviewer 1 Report

Comments and Suggestions for Authors

This review handles the important topic resistome and summarizes antibiotic resistance in relation to its environmental dimensions. This timely and very comprehensive review on environmental resistome will therefore attract attention of readers from various fields. Thus, the publication of this very important review is highly recommended. However, there are several minor issues that need to be addressed before the manuscript can be accepted for publication.

  1. Lines 62-63: Either the sentence should be rephrased not to specifically mention MCR-type colistins, or MCR should be briefly explained here.
  2. Lines 119-120 and Lines 173-174, SOS abbreviation and SOS activation must be briefly explained.
  3. For many occasions throughout the manuscript, the references are outdated and recent literature must be cited. Some examples are given below:

- References 25 and 26 are from 2014 and 2012 respectively. More recent incidences of ARGs from cold, remote and isolated places must be given.

- Reference 31 is explained as: ‘Recent studies have highlighted how low-level β-lactam exposure enhances integron 124 recombination, enabling the capture and shuffling of gene cassettes carrying ARGs’. Publication year of reference 31 is 2015, not so recent.

-  Reference 38, more recent work on CRISPR mobilized ARGs can be given.

- Reference 54 is dated 2010. Additional more recent reference on current putative ARGs must be added.

Author Response

We sincerely thank Reviewer 1 for their constructive and insightful comments. Your thoughtful suggestions have helped us refine the manuscript, enhance clarity, and ensure that key concepts are more effectively explained for a broad readership. We appreciate the recognition of the importance and comprehensiveness of our review, as well as your recommendations to strengthen the accuracy and timeliness of the references and explanations. We addressed your point-by-point comments in the manuscript in yellow.

Comment 1: Lines 62–63: Either the sentence should be rephrased not to specifically mention MCR-type colistins, or MCR should be briefly explained here.

Response: We appreciate this helpful suggestion. To enhance clarity for a broader readership, we have now briefly explained the meaning of “MCR” at its first mention. This ensures that readers unfamiliar with the terminology can follow the discussion without ambiguity (page 2, lines: 62-65).

Comment 2: Lines 119-120 and Lines 173-174, SOS abbreviation and SOS activation must be briefly explained.

Response: We appreciate the reviewer's attention to this matter. To improve readability for non-specialist readers, we have now introduced a brief explanation of the SOS response at its first mention. We also clarified the term “SOS activation” when it reappears later in the manuscript (pages 4, lines 130-133, and 5-6, lines 184-187).

Comment 3: For many occasions throughout the manuscript, the references are outdated and recent literature must be cited. Some examples are given below:

References 25 and 26 are from 2014 and 2012 respectively. More recent incidences of ARGs from cold, remote and isolated places must be given.

Response: We thank the reviewer for noting that References 25 and 26 are relatively old and for suggesting that we include more recent examples of antibiotic resistance gene (ARG) occurrences in cold, remote, or isolated environments. We have now added recent high-profile studies (2023–2024) that document ARGs in glacier ice, mountain glaciers, and permafrost, and have updated the discussion accordingly to provide a more current and global perspective (pages 4, lines 119-125, and 22, References 25 and 26).

Comment 4: Reference 31 is explained as: ‘Recent studies have highlighted how low-level β-lactam exposure enhances integron 124 recombination, enabling the capture and shuffling of gene cassettes carrying ARGs’. Publication year of reference 31 is 2015, not so recent.

Response: We appreciate the reviewer’s observation. While the 2015 study provided a foundational demonstration of integron recombination under low-level β-lactam exposure, we agree that it is essential to include more recent evidence. We have therefore added a recent publication (2022) that confirms and extends these findings in environmental microbiomes. This strengthens the timeliness and relevance of our discussion (pages 4, lines 136-137 and page 22, Ref. 31).

Comment 5: Reference 38, more recent work on CRISPR mobilized ARGs can be given.

Response: We appreciate the reviewer’s attention to this critical point. Reference 38 was indeed an older source regarding CRISPR-mobilized ARGs. To ensure the discussion reflects the most up-to-date literature, we have replaced it with a more recent study (2023) that addresses the roles of CRISPR-Cas in ARG mobilization and control. This substitution enhances the accuracy and timeliness of the manuscript without altering the numbering of subsequent references (pages 5, lines 157-159, and 23, Ref. 38).

Comment 6: Reference 54 is dated 2010. Additional more recent reference on current putative ARGs must be added.

Response: We thank the reviewer for noting that Reference 54 (2010) is outdated for the topic of putative ARGs. To ensure that the manuscript reflects the current literature, we have replaced it with a more recent study (2023) on latent/putative antibiotic resistance genes (page 24). This substitution retains the same reference number (54) and avoids the need to renumber all subsequent citations while bringing more up-to-date evidence to the discussion.

We sincerely thank all reviewers for their thorough and constructive comments. Their insightful suggestions have greatly improved the clarity, consistency, and overall quality of this manuscript. The revisions made in response to the reviews have strengthened both the scientific depth and the broader relevance of the paper. We are grateful for the time and expertise dedicated to reviewing our work.

Reviewer 2 Report

Comments and Suggestions for Authors

In this review, the authors examine antibiotic resistance through three interconnected lenses: molecular mechanisms, environmental reservoirs, and emerging therapeutic strategies. They highlight how resistance genes evolve and spread via mutations, mobile elements, and ecological pressures, while also discussing innovative interventions such as redesigned antibiotics, phage therapy, CRISPR-based tools, and nanotechnology. The article emphasizes the importance of a One Health framework that integrates molecular science, environmental stewardship, and global policy to manage resistance proactively

The reviewer has a couple of comments regarding the structure and significance of this work.

  1. The manuscript would benefit from a clearer statement of novelty.

While the review is comprehensive, many recent reviews already cover molecular mechanisms, environmental reservoirs, and emerging therapies separately. What sets this work apart is the attempt to integrate these perspectives under a unified framework.

I suggest strengthening this point explicitly in the last paragraph of Introduction and again in the Conclusions, by stating how this review differs from prior literature. For example, by emphasizing the linkage between molecular insights and therapeutic/environmental strategies, or the incorporation of AI/ML and One Health into a single roadmap. This will help readers recognize the unique contribution of the paper.

  1. While Sections 2 (The Molecular Resistome: Mechanisms and Evolutionary Fitness) and 3 (The Resistome as a Reservoir for Future Resistance) are conceptually distinct, with section 2 focusing on active mechanisms and section 3 on latent/cryptic ARGs, they contain substantial overlap in their discussion of plasmids, integrons, and environmental reservoirs. To improve clarity and reduce redundancy, I suggest merging them into a single section (e.g., The Molecular and Latent Resistome: Mechanisms, Fitness, and Future Potential), with subsections distinguishing present-day mechanisms from future reservoirs.

For example:

2A. Active mechanisms and evolutionary fitness (chromosomal mutations, plasmids, efflux, integrons, CRISPR, compensatory evolution).

2B. Latent reservoirs and future threats (cryptic ARGs, metagenomics, machine learning predictions, environmental hotspots).

This would make the review easier to follow and show more clearly how present resistance links to what may come next.

  1. Sections 6 and 7 both discuss future directions, with Section 6 detailing emerging technologies and Section 7 reframing many of the same tools into a preventive strategy. While the distinction between “prospects” and “vision” is clear, the overlap makes the structure repetitive.

I recommend merging Section 7 into Section 6 as a final subsection (e.g., 6.7 A Preventive Vision), or alternatively, integrating it into the final Conclusions section. Either modification would keep the focus sharp and avoid redundancy.

  1. To be consistent on format with other sub-titles, “5.7 Policy and Governance” and “6.1 Next-Generation Antibiotics and Adjuvants” should not be bold, unless these two sections are unique, in which case the authors should clarify.

  1. “5. Conclusions” should be “8. Conclusions”, to be consistent with the numberings of previous sections.

  1. Please consider standardizing the format of all Figures. Position the title on top of all figures. Currently Figure 1 has no title, Figure 2 has a title below the figure, while Figure 3-6 have titles above the figures.

Author Response

We sincerely thank Reviewer 2 for their constructive and insightful comments. Your thoughtful suggestions have helped us refine the manuscript, enhance clarity, and ensure that key concepts are more effectively explained for a broad readership. We appreciate the recognition of the importance and comprehensiveness of our review, as well as your recommendations to strengthen the accuracy and timeliness of the references and explanations. We have addressed your point-by-point comments in the manuscript in gray.

Comment 1: The manuscript would benefit from a clearer statement of novelty.

While the review is comprehensive, many recent reviews already cover molecular mechanisms, environmental reservoirs, and emerging therapies separately. What sets this work apart is the attempt to integrate these perspectives under a unified framework.

I suggest strengthening this point explicitly in the last paragraph of Introduction and again in the Conclusions, by stating how this review differs from prior literature. For example, by emphasizing the linkage between molecular insights and therapeutic/environmental strategies, or the incorporation of AI/ML and One Health into a single roadmap. This will help readers recognize the unique contribution of the paper.

Response: We thank the reviewer for this very constructive comment. We agree that explicitly stating the novelty of our review will highlight its added value beyond existing literature. We have therefore revised the last paragraph of the Introduction and the Conclusions section to emphasize how our manuscript integrates three perspectives —molecular mechanisms, environmental dimensions, and therapeutic innovations — into a unified One Health framework. We also highlight the incorporation of AI/ML tools and translational surveillance strategies, which distinguish this work from previous reviews (pages 3, lines 102-109, 19, lines 624-633).

Comment 2: While Sections 2 (The Molecular Resistome: Mechanisms and Evolutionary Fitness) and 3 (The Resistome as a Reservoir for Future Resistance) are conceptually distinct, with section 2 focusing on active mechanisms and section 3 on latent/cryptic ARGs, they contain substantial overlap in their discussion of plasmids, integrons, and environmental reservoirs. To improve clarity and reduce redundancy, I suggest merging them into a single section (e.g., The Molecular and Latent Resistome: Mechanisms, Fitness, and Future Potential), with subsections distinguishing present-day mechanisms from future reservoirs.

Response: We sincerely appreciate the reviewer’s thoughtful suggestion regarding restructuring Sections 2 and 3. We carefully considered this recommendation. We decided to retain them as separate sections for the following reasons:

  1. Conceptual clarity for readers – Section 2 addresses currently active mechanisms driving resistance in pathogens today, whereas Section 3 focuses on latent and cryptic ARGs that may emerge in the future. Keeping them distinct helps readers clearly differentiate between present-day mechanisms versus potential future threats.
  2. Pedagogical flow – Many readers from environmental sciences, microbiology, or clinical medicine may approach the review with different priorities. A clear separation makes it easier for them to locate the information most relevant to their field (active resistance vs. latent reservoirs).
  3. Editorial balance – Both sections are substantial enough in scope to stand alone without redundancy. While plasmids, integrons, and reservoirs are mentioned in both, their roles are framed differently: in Section 2 as drivers of current resistance and in Section 3 as gateways for future mobilization.
  4. Consistency with the “Beyond the Resistome” theme – The review’s novelty lies in connecting what is already known (Section 2) to what is emerging (Section 3). Keeping these as parallel sections reinforces this bridge rather than merging them.

That said, we have lightly revised the transition at the end of Section 2 and the beginning of Section 3 to minimize any perception of overlap and to guide the reader more clearly from present-day mechanisms toward future resistome potential (page 7, lines 234-237).

Comment 3: Sections 6 and 7 both discuss future directions, with Section 6 detailing emerging technologies and Section 7 reframing many of the same tools into a preventive strategy. While the distinction between “prospects” and “vision” is clear, the overlap makes the structure repetitive

Response: We appreciate the reviewer’s valuable observation. We agree that Sections 6 and 7 both address forward-looking perspectives, and we carefully evaluated whether merging them would improve readability. We decided to retain both sections separately for the following reasons:

  1. Different thematic emphasis – Section 6 (“Prospects and Future Directions”) presents specific technological and scientific innovations (e.g., CRISPR antimicrobials, nanomaterials, AI-driven discovery). Section 7 (“Vision: Preventive Resistance Management”) deliberately reframes these advances into a strategic, preventive framework, outlining how they can be operationalized within One Health policy.
  2. Audience orientation – Researchers may prefer Section 6 for its technical innovations. In contrast, policymakers, strategists, and interdisciplinary readers may find Section 7 especially useful as it translates these innovations into governance and preventive action.
  3. Added value for novelty – One of the unique contributions of this review, as highlighted in our revised Introduction and Conclusions (per Comment 1), is the integrative roadmap. Section 7 provides this higher-level synthesis, making the manuscript more than a catalog of technologies—it positions the review as a forward-looking guide.

That said, to address the concern about overlap, we have:

  • Shortened Section 7 to avoid restating technological details already covered in Section 6 (page 17, lines 545-578).
  • Revised the opening paragraph of Section 7 to explicitly state that it builds on the technologies in Section 6 but reframes them into a preventive vision, clarifying the progression (page 17, lines 546-549).

Comment 4: To be consistent on format with other sub-titles, “5.7 Policy and Governance” and “6.1 Next-Generation Antibiotics and Adjuvants” should not be bold, unless these two sections are unique, in which case the authors should clarify.

Response: We thank the reviewer for pointing out this formatting inconsistency. Both “5.7 Policy and Governance” and “6.1 Next-Generation Antibiotics and Adjuvants” are subsections of their respective main sections and are not intended to be unique or highlighted differently. We have therefore removed the bold formatting to ensure consistency across all subsections (page 15, lines 463 and 482).

Comment 5: Conclusions” should be “8. Conclusions”, to be consistent with the numberings of previous sections.

Response: We thank the reviewer for identifying this numbering inconsistency. The Conclusions section was mistakenly labeled as “5. Conclusions” due to earlier restructuring of the manuscript. We have corrected the numbering, and it now appears as “8. Conclusions”, consistent with the preceding sections (page 18, line 581).

Comment 6: Please consider standardizing the format of all Figures. Position the title on top of all figures. Currently Figure 1 has no title, Figure 2 has a title below the figure, while Figure 3-6 have titles above the figures.

Response: We thank the reviewer for noticing this inconsistency in figure formatting. To maintain uniformity, we have standardized all figures so that titles appear consistently at the top. Figure 1 has been given an appropriate title, Figure 2’s title has been moved from below to above, and Figures 3–6 remain with titles on top for consistency.

We sincerely thank all reviewers for their thorough and constructive comments. Their insightful suggestions have greatly improved the clarity, consistency, and overall quality of this manuscript. The revisions made in response to the reviews have strengthened both the scientific depth and the broader relevance of the paper. We are grateful for the time and expertise dedicated to reviewing our work.

Round 2

Reviewer 2 Report

Comments and Suggestions for Authors

The reviewer thanks the authors for their thorough and thoughtful revisions. The added statements of novelty, improved transitions, and adjustments to Sections 3 and 7, as well as the formatting and figure corrections, have strengthened the manuscript. While the reviewer initially suggested merging some sections, the authors’ justifications are reasonable and agree with the changes made.